# Record of low-temperature aqueous alteration of Martian zircon during the late Amazonian

Martin Guitreau [1,2] & Jessica Flahaut [3]

Several lines of evidence support the presence of liquid water on Mars at different times. Among those, hydrated minerals testify to past aqueous weathering processes that can be precisely studied in Martian meteorites such as NWA 7533/7034. Bringing constraints on the timing of weathering of the Martian crust would help understand its evolution, the availability of liquid water, and the habitability of Mars. Here we present a new method based on U–Th–Pb isotope systems to assess if zircon crystals underwent low-temperature aqueous alteration, such as exemplified by Hadean-aged detrital crystals from Western Australia. Data for NWA 7533 zircons show evidence for aqueous alteration and modeling of U–Th–Pb isotope system evolution indicates that the latest alteration event occurred during the late Amazonian (227–56 Ma). This finding largely expands the time duration over which liquid water was available near the Martian surface, thereby suggesting that Mars might still be habitable.

[1] School of Earth and Environmental Sciences, University of Manchester, Oxford road, Manchester M13 9PL, UK. [2] Université Clermont Auvergne, Laboratoire Magmas et Volcans, 6 avenue Blaise Pascal, 63178 Aubière, France. [3] CRPG, CNRS/Université de Lorraine, 54500 Vandœuvre-lès-Nancy, France. Correspondence and requests for materials should be addressed to M.G. (email: martin.guitreau@uca.fr)

Zircon is a very robust time-capsule that has been extensively used in U–Pb geochronology[1] and to study magmatic and metamorphic processes on Earth[2–4]. Zircon also has the capacity to survive the destruction of its host-rock and to be deposited in detrital sediments, hence, becoming an archive of rock formations. This is testified to by the terrestrial detrital zircon record that goes back to 4378 million years[5]. However, the strength of zircon as a geochronometer is also its Achille's heel. Alpha-particle emissions and α-recoil cascades due to U and Th radioactive decay chains damage the crystal lattice[6,7]. Radiation damage accumulates in zircon at different rates, depending on U and Th concentrations, until amorphous domains become connected. This stage is called the first percolation point[8,9] and once it is reached, chemical elements can be mobilized more readily than in pristine crystals. The increase of amorphization induces lattice expansion[6,10], which results in the formation of cracks in zircon[10,11]. This effect enhances zircon's sensitivity to thermal events and external fluid infiltrations as evidenced by modification of O isotope signatures[12,13], discordance of U–Pb ages[10,13,14], and enrichment in non-formula elements (e.g., Ca, Al, K, Fe)[15]. These features of radiation-damaged zircon were used to argue for hydrothermal alteration of 4.43 Gyr-old Martian zircons by Nemchin et al.[16] and McCubbin et al.[17].

It was recently demonstrated[13] that some Hadean-aged zircon crystals from the Jack Hills supracrustal belt (Australia) showed elevated O isotope signatures, measurable amounts of OH radicals, and discordant U–Pb ages. Zircon is nominally anhydrous and the presence of OH is interpreted as the result of low-temperature weathering (≤50 °C) by aqueous solutions, since at least the Permian[13]. Jack Hills zircons locally show advanced signs of radiation damage, such as domains with weak cathodoluminescence signal, abundant cracks, and spongy textures[13,18,19]. Pidgeon et al.[13] concluded that aqueous solutions infiltrated within zircons through cracks and into amorphous domains, which resulted in local modifications of O isotope signatures, deposition of OH radicals and increase of U and Th concentrations. These features were essentially detected once analytical spots overlapped fractures. Therefore, these authors suggested that the zero-age U–Pb discordance exhibited by the Jack Hills zircons was due to U and Th gain during weathering. These Hadean crystals further exhibit a wide range of measured Th/U with values up to 5.5, which largely falls outside the common magmatic range (~0.2–1; average = 0.5)[20]. There are essentially two ways to determine Th/U in zircon using mass spectrometry. It can be obtained directly by measuring Th and U abundances or it can be calculated from the decay products of $^{232}$Th and $^{238}$U (i.e., $^{208}$Pb and $^{206}$Pb[21]), the latter being referred to as the time-integrated Th/U. In the vast majority of cases, measured and time-integrated Th/U agree fairly well, as exemplified by data in Fig. 1a, which is a compilation of oldest known terrestrial igneous zircons. In contrast, data for Hadean-aged Jack Hills zircon grains consistently show a horizontal distribution in Fig. 1b. The large range of measured Th/U (0.2–5.5) exhibited by Pidgeon et al.[13] dataset have associated time-integrated Th/U that fall exactly within the common magmatic range, hence, clearly decoupled from each other. Figure 1b further shows that this kind of decoupling is not dataset-dependent but is an actual feature of these ancient detrital zircon crystals and arise from the low-temperature aqueous alteration described in Pidgeon et al.[13]. The term decoupling is not referred to as a mechanism but merely used to describe the fact that measured and time-integrated Th/U are different.

In the present contribution, we explore the possibility that decoupling between measured and time-integrated Th/U in zircon can be used as a proxy for low-temperature aqueous alteration and outline the principles of this new method. When applied

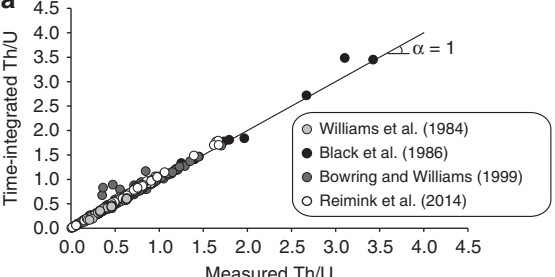

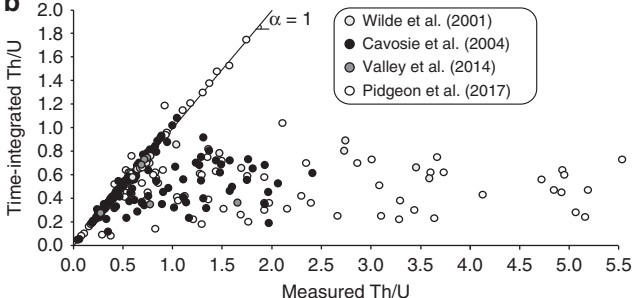

**Fig. 1** Comparison of measured and time-integrated Th/U in zircon. The panel **a** is for oldest known terrestrial igneous zircons[54–57], whereas panel **b** is for oldest known terrestrial detrital zircons from Western Australia[5,13,58,59]. Time-integrated means calculated from measured $^{208}$Pb/$^{206}$Pb ratio and $^{207}$Pb/$^{206}$Pb age. Corresponding data can be found in Supplementary Data 1 and 2

to extraterrestrial zircons, our method suggests the absence of aqueous alteration on the Moon but, in contrast, reveals that Martian zircons show evidence for low-temperature aqueous weathering. We further model the evolution of the U–Th–Pb isotope systems in zircon to determine the timing of development of decoupling between measured and time-integrated Th/U, and demonstrate that although alteration did occur at 1500–1700 Ma, a much younger event is required to account for the data. This event corresponds to the late Amazonian which is a period generally considered cold and dry on Mars. As a consequence, our results demonstrate that liquid water was available near the Martian surface in a recent past and might still be in the present-day.

## Results

**U–Th–Pb isotopes as a proxy for aqueous weathering**. An important aspect to consider when one wants to assess the sensitivity of zircon to chemical modifications and/or isotopic resetting is the preservation of its lattice. This can be assessed at a macroscopic scale using visual criteria (e.g., textures revealed by CL and BSE images) and at a microscopic scale by several techniques, which include, for example, transmission electron microscopy[9,22], Raman spectroscopy[23], X-ray diffraction[9,24], nuclear magnetic resonance[25], and more recently atom-probe tomography[26]. A simpler approach, which is however not a direct observation, is to calculate the radiation dose (i.e., alpha-decay events per gram of sample) that a zircon underwent using chronological information provided by U–Pb isotope systems as well as U and Th concentrations[6,9]. Previous studies have shown general good agreements between these calculated radiation doses and the observed crystallinity in natural and artificial zircons[6,8,25]. However, zircon lattice can heal during thermal events and radiation doses are, hence, only indicative[27]. Measured and time-integrated Th/U presented as a function of calculated radiation doses for Jack Hills zircon datasets (Fig. 2) demonstrate that the

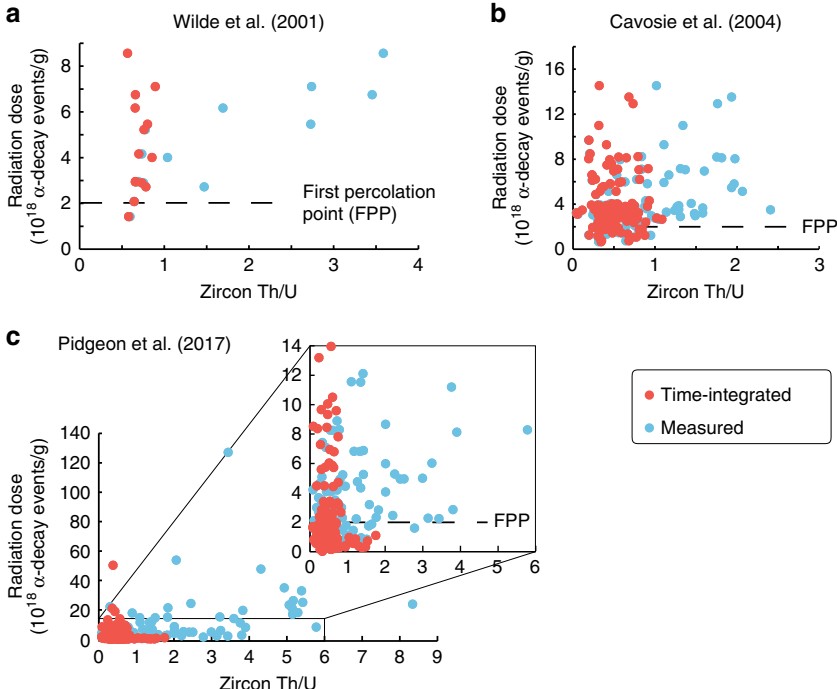

**Fig. 2** Calculated radiation doses as a function of Th/U in zircon. The panels **a**, **b** and **c** correspond to data from Wilde et al.[58], Cavosie et al.[59], and Pidgeon et al.[13], respectively. Note the general upward-opening fan shape distribution of data above the first percolation point (stage at which amorphous domains become connected[8,9]). Corresponding data can be found in Supplementary Data 2

decoupling between measured and time-integrated Th/U occurred in zircon domains that accumulate radiation doses beyond the first percolation point (i.e., $2 \times 10^{18}$ alpha-decay events/g[8]). This observation strongly supports Pidgeon et al.[13] conclusion that the decoupling between measured and time-integrated Th/U were due to penetration of aqueous solutions, because the formation of cracks in zircon is mostly controlled by radiation damage accumulation. It is tempting to use this decoupling as a proxy for low-temperature aqueous alteration of ancient zircon crystals, and especially extra-terrestrial ones, to search for evidence for liquid water outside the Earth. It is, nevertheless, relevant to inquire if other processes could decouple measured and time-integrated Th/U prior to drawing conclusions on aqueous alteration outside the Earth.

Lead mobility which accounts for some cases of U–Pb age discordance can decouple Pb from U and Th, thereby altering the final Pb isotope composition of a crystal[1,28]. However, half-lives for $^{232}$Th and $^{238}$U are so long that over the Earth's history the factor relating $^{208}$Pb/$^{206}$Pb to Th/U varies from 0.25 to 0.32[21]. This translates into a 6% maximum increase of the time-integrated Th/U for a 4430 Ma zircon that lost 90% of its Pb (Fig. 3). This is not significant compared to the range of Th/U exhibited by natural zircons[20,21,29] and correspond to the opposite behavior to that exhibited by terrestrial Hadean zircons. A second possibility is that a zircon Th/U is modified during metamorphism[3,29]. However, high Th/U in metamorphic zircon is generally achieved in granulite facies which is incompatible with the temperature of metamorphism described for the martian meteorite NWA 7533 and paired stones (including NWA 7034)[17,30]. Furthermore, ratios as high as those exhibited by weathered crystals in Fig. 1b are extremely rare in metamorphic zircons. Change of Th/U during metamorphism is most likely achieved by dissolution–reprecipitation processes, which would therefore expel most of the Pb from the newly crystallized zircon as it is essentially incompatible within its lattice[31]. Therefore, without memory of past radiogenic Pb

accumulation, no decoupling can be recorded. A third option for decoupling measured and time-integrated Th/U would consist in adding common Pb to the lattice. Common Pb has high $^{208}$Pb/$^{206}$Pb that varied from 3.4 to 2 over the Earth's history[32], which would translate into time-integrated Th/U of 13.8–6.6. Consequently, the presence of common-Pb would increase the time-integrated Th/U, hence, creating a vertical distribution in diagrams such as Fig. 1. It should also be noted that $^{204}$Pb, hence common Pb, is routinely monitored and corrected for in SIMS measurements. We, therefore, conclude that the decoupling between measured and time-integrated Th/U exhibited by terrestrial zircons can be used as a proxy for zircon alteration by aqueous solutions at low temperature.

**Evidence for aqueous weathering on Mars**. A compilation of available U–Th–Pb isotope data for lunar zircons is presented in Fig. 4 and Supplementary Data 3, and these data do not show actual decoupling between measured and time-integrated Th/U, except due to common Pb, despite some radiation doses exceeding that of the first percolation point. Moreover, lunar zircons do not exhibit anomalously high Th/U but on the contrary comply with the common range for terrestrial igneous zircons[20]. This is expected because there is no evidence for liquid water on the Moon[33–35]. Martian zircon crystals found in monzonitic clasts from NWA 7533[16,36], as well as in protobreccia clasts and the matrix of NWA 7034[17] also exhibit very consistent measured and time-integrated Th/U (Fig. 5). The only datapoint that falls out of the 1:1 correlation is due to common Pb. In contrast, Nemchin et al.[16] data for Martian zircons from NWA 7533 show at the same time good correlation, vertical distribution, and horizontal distribution (Fig. 5). We interpret the horizontal distribution as evidence for low-temperature alteration (≤50 °C) of Martian zircon grains by aqueous solutions much as what was experienced by crystals from Western Australia. Nemchin et al.[16] interpreted Martian zircon O isotope compositions as evidence for interaction with the Martian hydrosphere,

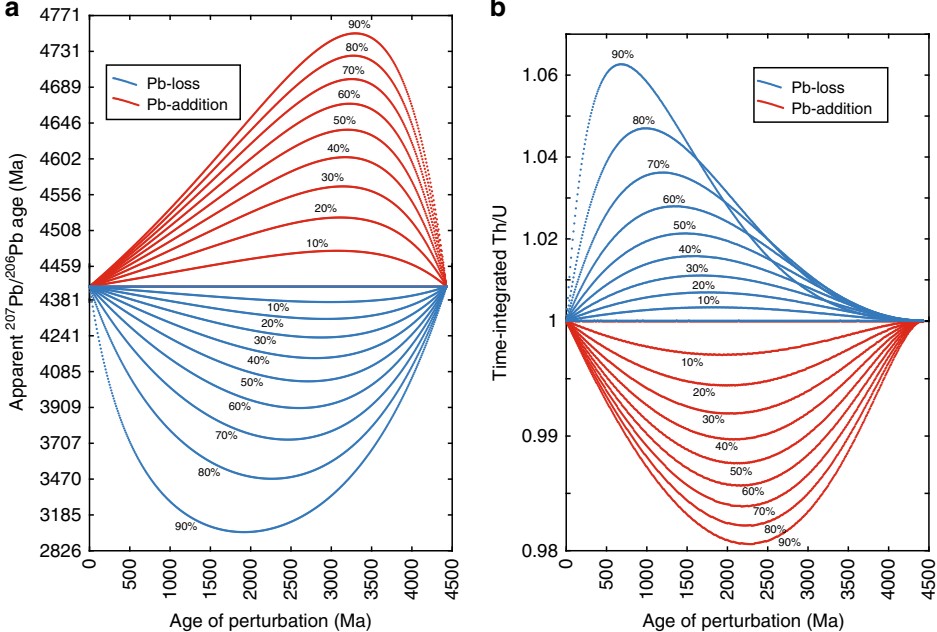

**Fig. 3** Influence of Pb mobility on present-day $^{207}Pb/^{206}Pb$ ages and time-integrated Th/U. The model used to build these panels is described in the "Methods" section. The panel (**a**) shows the effect of Pb-loss and Pb-addition on determined $^{207}Pb/^{206}Pb$ ages for a 4430 Ma zircon with original Th/U of 1, whereas the panel (**b**) presents the effect of Pb-loss and Pb-addition on time-integrated Th/U for the same zircon. Percentages next to the dotted-curves correspond to the degree of Pb-loss or Pb-addition. The age of perturbation refers to the age at which the Pb is lost or added. Note that Pb-addition corresponds to local increase in radiogenic Pb concentration and not addition of common Pb. This figure illustrates the great sensitivity of $^{207}Pb/^{206}Pb$ ages to Pb mobility whereas it has a limited effect on time-integrated Th/U

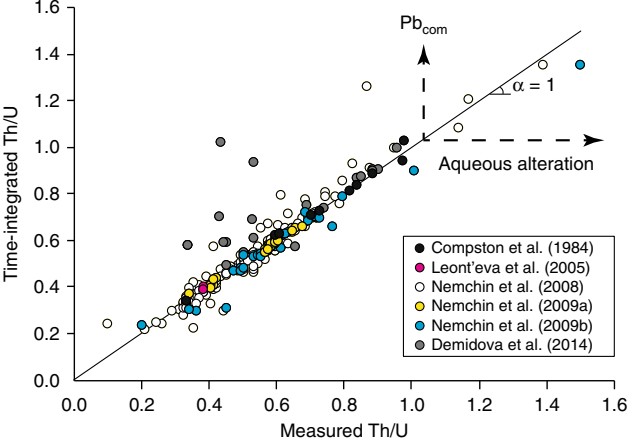

**Fig. 4** Time-integrated Th/U as a function of that measured for lunar zircons. Corresponding data are presented in Supplementary Data 3 and come from Compston et al.[60], Demidova et al.[61], Leont'eva et al.[62], and Nemchin et al.[63–65]. Note the general good agreement between measured and time-integrated Th/U, as well as the fact that Th/U range between 0.2 and 1, which is identical to terrestrial igneous zircons[20]

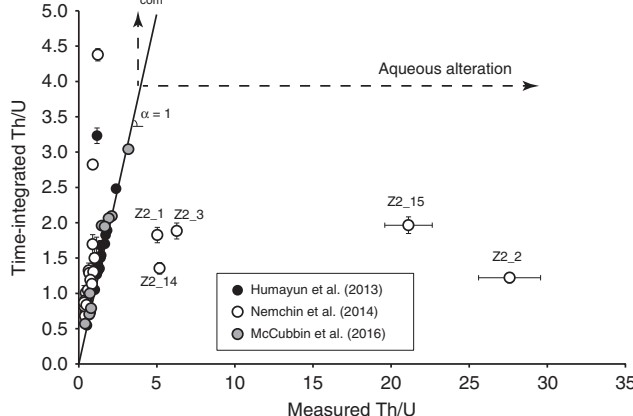

**Fig. 5** Time-integrated Th/U as a function of that measured for Martian zircons. Data are from Nemchin et al.[16], McCubbin et al.[17], and Humayun et al.[36] which are available in Supplementary Data 4. Note the large decoupling between measured and time-integrated Th/U for domains within zircon Z2 from NWA 7533 analyzed in Nemchin et al.[16]. Measured Th/U are also well outside the common magmatic range, much as what can be seen in Jack Hills zircons. Error bars represent two standard errors on analytical measurements

and McCubbin et al.[17] interpreted some U-rich Martian zircons to have been hydrothermally altered because of their elevated $K_2O$ content (>1 wt%), which zircon is normally devoid of. Finally, Liu et al.[37] showed Nemchin et al.[16] and McCubbin et al.[17] findings to be consistent with observed spongy zircon textures as revealed by CL images for NWA7034/7533 zircons. Therefore, our finding using decoupling between measured and time-integrated Th/U reinforces the idea that liquid water was available in the Martian sub-surface to induce advanced weathering of radiation-damaged zircon crystals.

**Timings of Martian aqueous weathering**. Some phases in NWA7034, and paired meteorites, arguably experienced aqueous alteration[38–40]. Unraveling the timing of this alteration is complicated by the fact that NWA 7034 and 7533 are Martian regolith breccias that contain proto-breccia clasts with textural evidence for multiple alteration events[17,38–40]. Nemchin et al.[16] suggested that zircon alteration in NWA 7533 occurred at 1700 Ma, as indicated by the U–Pb Discordia lower-intercept[36]. McCubbin et al.[17] obtained a lower-intercept at 1500 Ma for NWA 7034

zircons, which is consistent with NWA 7034 baddeleyite and apatite age of 1500 Ma. McCubbin et al.[17] interpreted this age to date a thermal event linked to NWA7034 formation as a breccia, and further suggested that zircon alteration occurred prior to this event because some magnetite veins in the proto-breccia, from which most zircons were analyzed, showed evidence for pre-NWA7034 alteration. In fact, petrographic description of secondary mineral distribution indicates that some clasts dated at 1000 Ma show evidence for alteration prior to incorporation in NWA 7034 and paired stones, whereas some younger alteration veins clearly crosscut the matrix[38–40]. A recent study by Cassata et al.[30] further suggested that brecciation of NWA 7034 occurred much later than previously thought, around 225 Ma. We developed a two-stage model of U–Th–Pb isotope evolution to test if an alteration event that occurred at 1500 or 1700 Ma can account for the observed decoupling between measured and time-integrated Th/U (Fig. 5), and, hence, that of zircon alteration in NWA 7533 (see the section "Methods"). The first stage in our model corresponds to zircon formation at 4430 Ma, and the second stage is an alteration event that consists in increasing U and Th concentrations, as well as decoupling Th from U so as to obtain Th/U ratios that equal those observed in Fig. 5. We imposed that uranium could be enriched by 2–5 times the original concentration so as to match both Jack Hills and Martian zircon data (Supplementary Data 2 and 4). McCubbin et al.[17] data clearly show a dichotomy between ancient, well-behaved zircons with low U and Th contents, and younger, altered crystals with very high U and Th concentrations. This dichotomy is also expressed in measured Th/U ratios because altered zircons have Th/U between 1.5 and 3, whereas ancient well-behaved zircons

have Th/U between 0.4 and 0.8 (Supplementary Data 4), consistent with terrestrial igneous zircons[20], unaltered lunar zircons (Fig. 4), and Martian zircons analyzed by ID-TIMS[41], which have Th/U between 0.6 and 0.7. Consequently, we think that original (pre-alteration) Th/U ratios of 0.5–1 bracket the natural variability observed in igneous zircons quite well, and we have, hence, used these values in our model. The results of our two-stage model are presented in Fig. 6 and show that the decoupling between measured and time-integrated Th/U visible in Fig. 5 cannot have occurred at 1700 Ma, nor at 1500 Ma, because obtained time-integrated Th/U are higher than those in zircon Z2 domains. This means that the decoupling and, hence, the alteration event must have occurred much more recently than 1500–1700 Ma (Fig. 6). Yet, these results provide an answer to the apparent contradiction that zircon crystals analyzed by McCubbin et al.[17] do not show decoupling between measured and time-integrated Th/U despite ample evidence for alteration. The alteration was, indeed, too old and not large enough (Th/U < 3) so that $^{232}Th/^{238}U$ and $^{208}Pb/^{206}Pb$ returned to secular equilibrium after decoupling, much like what U-series nuclides do[42]. The two-stage model having failed to reproduce the decoupling observed in Fig. 5, we further implemented our model with an additional stage so as to account for igneous crystallization of zircon at 4430 Ma, metamorphism at 1500–1700 Ma, and recent alteration (see the section "Methods"). In this three-stage model, we have used the same Th/U ratios and uranium enrichment factors as in the two-stage model. We also tested the influence of the age of the thermal event (i.e., 1700 and 1500 Ma) so as to give credit to both Humayun et al.[36] and McCubbin et al.[17] datasets. Finally, for a fixed set of parameters we have averaged alteration ages in order

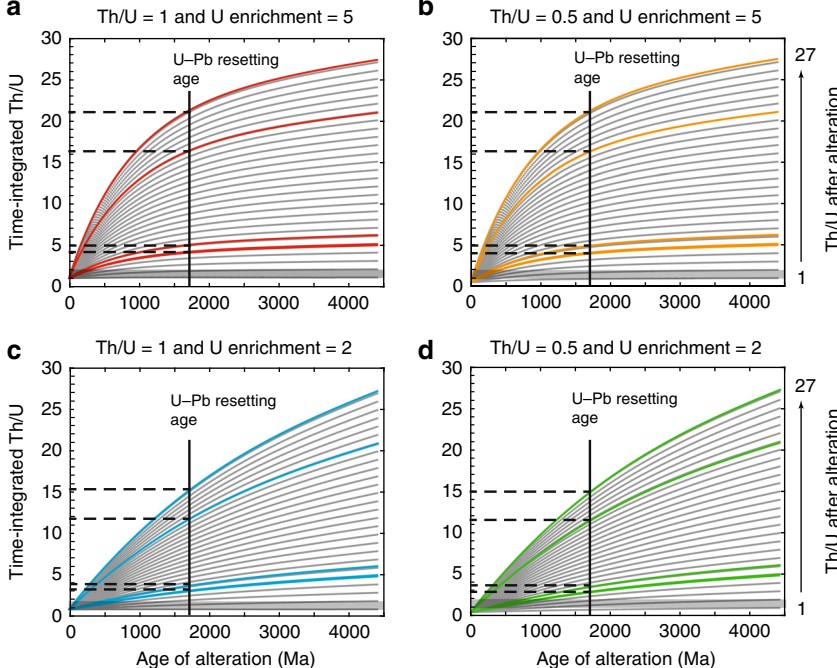

**Fig. 6** Results for the two-stage model simulating Martian zircon evolution (see the section "Methods" for details). Models were run for U enrichment factors of 5 for panels **a** and **b**, whereas a factor of 2 was used for the results presented in panels **c** and **d**. The pre-alteration Th/U was set to 1 for models shown in panels **a** and **c**, and it was set to 0.5 in panels **b** and **d**. Th/U ratios increased by alteration were set to 1–27 and 0.5–27, depending on the pre-alteration Th/U. Colored curves correspond to measured Th/U in zircon Z2 domains (Supplementary Data 4). The U–Pb resetting age corresponds to the lower intercept of the general Discordia line displayed by Martian zircons in Humayun et al.[36] and is represented by a solid vertical line. The horizontal dashed lines, which are the intercepts between the colored curves and the solid vertical line, indicate the time-integrated Th/U deduced from our model. The gray-shaded zones between Th/U values of 1 and 2 correspond to the time-integrated Th/U exhibited by zircon Z2 domains (Supplementary Data 4). These results show that time-integrated Th/U derived from our model are higher than those exhibited by zircon Z2 domains, and, hence, alteration at 1700 Ma (or 1500 Ma) cannot account for the observed decoupling between measured and time-integrated Th/U in Fig. 5

**Table 1 Results for the three-stage model applied to zircon Z2**

| Domain | Th/U$_{orig}$ = 0.5 and $\phi$UE = 2 Age of alteration (Ma) | Th/U$_{orig}$ = 0.5 and $\phi$UE = 5 Age of alteration (Ma) | Th/U$_{orig}$ = 1 and $\phi$UE = 2 Age of alteration (Ma) | Th/U$_{orig}$ = 1 and $\phi$UE = 5 Age of alteration (Ma) |
|---|---|---|---|---|
| Z2_1 | 314 | 139 | 209 | 88 |
| Z2_2 | 67 | 27 | 21 | 9 |
| Z2_3 | 261 | 112 | 174 | 73 |
| Z2_14 | 312 | 131 | 150 | 61 |
| Z2_15 | 181 | 74 | 121 | 49 |
| Avg. alteration age | 227 | 97 | 135 | 56 |

Th/U$_{orig}$ stands for the pre-alteration Th/U of the zircon and $\phi$UE is the uranium enrichment factor of the zircon due to alteration

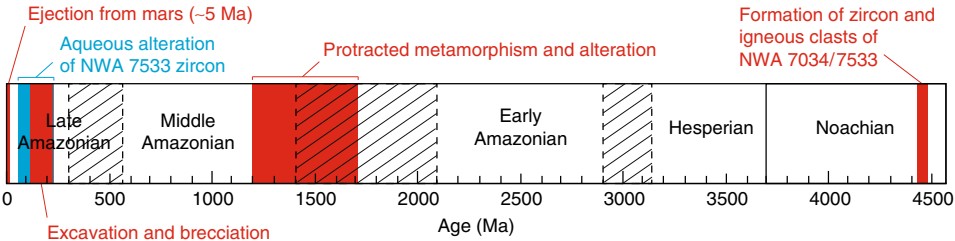

**Fig. 7** Timeframe for Martian crust evolution[66]. Displayed are details of knowledge about the history of NWA 7034/7533 and paired stones. The igneous crystallization of NWA7034/7533 zircons and clasts are derived from McCubbin et al.[17], Humayun et al.[36], Nyquist et al.[67], Tartèse et al.[68], and Yin et al[69]. Metamorphism and alteration ages are from McCubbin et al.[17], Cassata et al.[30], Humayun et al.[36], Liu et al.[39], Cartwright et al.[43], Tartèse et al.[68], and Lindsay et al.[70]. Brecciation age range is from Cassata et al.[30] and Cartwright et al.[43], and so is the age of ejection

to obtain more robust estimates. Results presented in Table 1 demonstrate that the age of the metamorphic event has no bearing on the alteration ages (Supplementary Data 5), in contrast to pre-alteration Th/U and uranium enrichment factors (Table 1). However, all determined alteration ages remain very young because they range from 56 to 227 Ma (Table 1). The alteration event recorded by zircon Z2 domains, therefore, occurred during the late Amazonian (Fig. 7).

## Discussion

Although, Pidgeon et al.[13] suggested that an extensive and deep weathering affected Jack Hills zircon crystals, they also acknowledged that the movement and reaction of low-temperature fluids in zircon are presently poorly understood. It is, consequently, unclear if the range of ages obtained with our model corresponds to the duration of an alteration event, or represents multiple unresolvable events, or is an artifact of modeling, or a combination of both. Nevertheless, these results account for zircon data that have ~zero-age U–Pb discordance and which suggest post-1500 Ma disturbance[17,30,36]. More importantly, zircon alteration ages are very consistent with U–Th–He and U–Th–Sm–He ages of 170 Ma and 113–135 Ma[30,43] that led Cassata et al.[30] to argue that excavation and brecciation of NWA 7034 and paired stones occurred at 225 Ma or later. Brecciation can, in fact, create porosity in which free-water is able to percolate, access minerals and, hence, alter them. Petrographic and geochemical evidence argue for pre-brecciation and post-brecciation alteration in NWA 7034 and paired stones[16,17,36–38], and we, therefore, propose that the ages obtained with our model date the post-brecciation pervasive alteration that is visible in the matrix of these meteorites[38]. A critical aspect to evaluate in order to validate the analogy between weathering processes on Mars and on the Earth is the redox and pH characteristics of the aqueous solutions as reduced form of U is insoluble[44]. According to Lorand et al.[38] pyrite crystals deposited during hydrothermal alteration of NWA 7533 were formed under oxidizing conditions and just-saturated to near

neutral pH, hence in conditions alike those on Earth. Although the timing does not match, one can wonder if zircon alteration did not occur on Earth after the fall of NWA 7533. Despite evidence for some (limited) terrestrial alteration in meteorites found in hot deserts[38], we estimate that it cannot account for the decoupling between measured and time-integrated Th/U because O isotope data for Z2 zircon coherently align along the Martian fractionation line[16], in addition to being consistent with carbonates from Martian meteorite ALH 84001[16,45]. Moreover, hot desert environments are not likely to provide enough liquid water to allow deep weathering, such as that experienced by Martian zircons, especially because if it were the case, NWA 7034/7533 would contain abundant calcite and barite veins typical of terrestrial alteration, which they do not.

The low-temperature alteration event recorded in NWA 7533 zircons occurred during the late Amazonian, which is a period generally considered cold and dry[46,47]. However, recent orbital observations point to the existence of a few early to mid-Amazonian fluvial features[47] and evidence has been reported for Amazonian alteration in SNC meteorites. This latter is substantiated by iddingsite in the Lafayette Nakhlite dated at 670 ± 91 Ma[48], and by carbonates and sulfates that occur in shergottites that have (debated) igneous crystallization ages of <475 Ma[49]. The availability of water for weathering in the late Amazonian was likely controlled by impact-induced hydro-thermal activity, consistent with post-brecciation zircon altera-tion, but the youngest volcanic activity on Mars[47] can also have played a role by interaction with the current cryosphere. Altera-tion of zircons in NWA 7533 represents the youngest episode of pervasive aqueous alteration reported so far for Mars surface. This result supports the idea that Mars may still be habitable in the present-day as argued by the availability of liquid water in a recent past.

## Methods

**Modeling of zircon lattice damage by alpha-decay.** The radiation dose that has accumulated in a zircon can be calculated using the following equation taken from

Murakami et al.[9] and updated from Holland and Gottfried[7]:

$$D_\alpha = 8N_{238} \times (e^{\lambda_{238}t} - 1) + 7N_{235} \times (e^{\lambda_{235}t} - 1) + 6N_{232} \times (e^{\lambda_{232}t} - 1) \quad (1)$$

where $D\alpha$ is the dose in α-decay events per gram, $N_{238}$, $N_{235}$, and $N_{232}$ the respective number of atoms of $^{238}U$, $^{235}U$, and $^{232}Th$ per gram, $\lambda_{238}$, $\lambda_{235}$, and $\lambda_{232}$ the decay constants of $^{238}U$, $^{235}U$, and $^{232}Th$, respectively, and $t$ the age (e.g., $^{207}Pb/^{206}Pb$ age).

Commonly agreed upon decay constants for $^{238}U$, $^{235}U$, and $^{232}Th$ are $1.55125 \times 10^{-10}$, $9.8485 \times 10^{-10}$, and $4.9475 \times 10^{-11}$, respectively[50,51]. The present-day $^{238}U/^{235}U$ ratio is 137.818[52].

**Modeling of U–Th–Pb isotope systematics evolution in zircon.** The end product of Pb, U, and Th evolution in zircon can be calculated using the following classical equations:

$$P_t = P_0 \times e^{\lambda_p t} \quad (2)$$

$$D_t = D_{in} + P_0 \times (e^{\lambda_p t} - 1) \quad (3)$$

where $P$ is the quantity of radioactive parent isotopes (i.e., $^{238}U$, $^{235}U$, and $^{232}Th$), $D$ that of the radiogenic daughter isotopes (i.e., $^{206}Pb$, $^{207}Pb$, and $^{208}Pb$), and $\lambda_p$ the decay constant of the specific parent. Subscripts $t$, $_0$, and $_{in}$ refer to anytime, present-day, and time of zircon formation, respectively.

A time-integrated Th/U value calculated from Pb isotopes can be determined using the following equation:

$$\left(\frac{^{232}Th}{^{238}U}\right)_t = \left(\frac{^{208}Pb}{^{206}Pb}\right)_t \times \left(\frac{e^{\lambda_{238}t} - 1}{e^{\lambda_{232}t} - 1}\right) \quad (4)$$

where $t$ is either the crystallization age or an apparent $^{207}Pb/^{206}Pb$ age.

**Lead mobility.** In the case where a zircon underwent punctual Pb-loss, at some stage in its evolution, the final daughter isotope quantity can be determined using the following equation:

$$D_t = D_{in} + P_0 \times (e^{\lambda_p t_1} - e^{\lambda_p t_2}) \times (1 - \varphi_{loss}) + P_0 \times (e^{\lambda_p t_2} - 1) \quad (5)$$

where $t_1$ and $t_2$ represent time of zircon crystallization and time of Pb-loss, respectively, and $\varphi_{loss}$ refers to the percentage of Pb lost during a distinct thermal event.

In the case of local Pb addition, possibly due to redistribution of Pb within zircon lattice[53], the equation becomes:

$$D_t = D_{in} + P_0 \times (e^{\lambda_p t_1} - e^{\lambda_p t_2}) \times (1 + \varphi_{add}) + P_0 \times (e^{\lambda_p t_2} - 1) \quad (6)$$

where $t_1$ and $t_2$ represent time of zircon crystallization and time of Pb-addition, respectively, and $\varphi_{add}$ refers to the percentage of Pb locally added to a zircon domain. Note that for the two above-mentioned scenarios, U and Th concentrations within zircon are modified only by radioactive decay over a long period of time owing to their long half-lives. Equations (5) and (6) have been used to obtain data presented in Fig. 1.

**Uranium and thorium concentration increase.** If instead of having Pb mobilized a zircon gains U and/or Th, much like what was experienced by Jack Hills zircons[13], we can introduce an enrichment factor $\phi_{UE}$ and the quantity of Pb can be modeled using this equation:

$$D_t = D_{in} + P_{t1} \times (e^{\lambda_p t_1} - e^{\lambda_p t_2}) + P_{t1} \times \phi_{UE} \times (e^{\lambda_p t_2} - 1) \quad (7)$$

where $t_1$ and $t_2$ represent time of zircon crystallization and time of U and/or Th enrichment. $P_{t1}$ is the abundance of either $^{232}Th$ or $^{238}U$ prior to the enrichment.

If we now assume that Th enrichment is larger than that of U, alike in altered zircons, we can set two specific enrichment factors that are linked by the following equation:

$$\phi_{ThE} = \phi_{UE} \times \frac{Th}{U} \quad (8)$$

with $\phi_{UE}$ and $\phi_{ThE}$, the U and Th enrichment factors, respectively, and Th and $U$, the Th and U concentrations observed in a zircon.

We have, now, to write three different equations for U and Th from Eq. (7):

$$^{206}Pb_t = {}^{206}Pb_{in} + {}^{238}U_{t1} \times (e^{\lambda_{238}t_1} - e^{\lambda_{238}t_2}) + {}^{238}U_{t1} \times \phi_{UE} \times (e^{\lambda_{238}t_2} - 1) \quad (9)$$

$$^{207}Pb_t = {}^{207}Pb_{in} + {}^{235}U_{t1} \times (e^{\lambda_{235}t_1} - e^{\lambda_{235}t_2}) + {}^{235}U_{t1} \times \phi_{UE} \times (e^{\lambda_{235}t_2} - 1) \quad (10)$$

$$^{208}Pb_t = {}^{208}Pb_{in} + {}^{232}Th_{t1} \times (e^{\lambda_{232}t_1} - e^{\lambda_{232}t_2}) + {}^{232}Th_{t1} \times \phi_{ThE} \times (e^{\lambda_{232}t_2} - 1) \quad (11)$$

**Two-stage model of U–Th–Pb isotope evolution in zircon.** If we assume that Martian zircons experienced a two-stage history, with crystallization at 4430 Ma and alteration at 1700–1500 Ma, such as suggested by Nemchin et al.[16] and McCubbin et al.[17], we can use Eqs. (9)–(11) to test this hypothesis. We ignore the

pre-alteration Th/U of Martian zircons but we can reasonably estimate that they fall in the range 0.2–1, much like terrestrial igneous zircons[20] and lunar zircons (Fig. 4; Supplementary Data 3). McCubbin et al.[17] data give a good estimate of pre-alteration Th/U because the oldest and most concordant crystals with low U and Th have Th/U ranging from 0.4 to 0.8. Moreover, time-integrated Th/U for NWA 7034 zircons with concordant 4430 Ma ages and igneous REE patterns analyzed by ID-TIMS in Bouvier et al.[41] have values between 0.6 and 0.7, hence, very consistent with above-mentioned values. On the contrary, altered zircons that have young ages and high U and Th concentrations in McCubbin et al.[17] data display Th/U between 1.5 and 3. We, therefore, think that values of 0.5 and 1 brackets well the possibilities for pre-altered Th/U in Martian zircons. The U enrichment factor linked to the alteration event was set to 2 and 5 in order to account for the fact that most enriched crystals have U concentrations on the order of 1500 ppm (Supplementary Data 4), and that in order to have accumulated enough radiation damage at 1500–1700 Ma, zircons must have contained at least 300 ppm. Results for this two-stage model are presented in Fig. 6 and discussed in the main text.

**Three-stage model of U–Th–Pb isotope evolution in zircon.** In order to evaluate the alteration age of Martian zircon Z2[16], which shows a decoupling between measured and time-integrated Th/U, we developed a three-stage model that simulates igneous crystallization, then, metamorphism at 1500–1700 (i.e., Pb-loss) and, finally, alteration (i.e., increase of U and Th concentrations). Consequently, we have introduced a Pb-loss term (Eq. (5)) to simulate the thermal event (i.e., 1500–1700 Ma) within Eq. (7), which hence becomes:

$$D_t = \left[D_{in} + P_{t1} \times (e^{\lambda_p t_1} - e^{\lambda_p t_2})\right] \times (1 - \varphi_{loss}) + P_{t1} \times (e^{\lambda_p t_2} - e^{\lambda_p t_3})$$
$$+ P_{t1} \times \phi_{UE} \times (e^{\lambda_p t_3} - 1) \quad (12)$$

where $t_1$, $t_2$, and $t_3$ represent time of crystallization, time of metamorphism and time of alteration, respectively.

We have set $\phi_{UE}$ to 2 and 5, and pre-altered Th/U to 0.5 and 1. The $\phi_{ThE}$ was adapted so as to reproduce measured Th/U in zircon Z2 domains. Given that Z2 data fall on the general Discordia line between 4428 and 1712 Ma presented by Humayun et al.[36], we interpret, much as these authors do, the former and the latter dates as reflecting igneous crystallization and disturbance of the U–Pb system, respectively. Nemchin et al.[16] only report $^{207}Pb/^{206}Pb$ ratios, hence apparent ages, for NWA 7533 zircon crystals, which in principle prevents $\varphi_{loss}$ to be assessed. However, because Humayun et al.[36] showed that NWA 7533 zircons have U–Pb systematics that fall on a single Discordia line, it is possible to retrieve $^{206}Pb/^{238}U$ and $^{207}Pb/^{235}U$ values for each domain in Z2 by using reported $^{207}Pb/^{206}Pb$ ages and the coordinates of the 4430–1700 Ma Discordia line. Because the apparent ages for zircon domains Z2_1 and Z2_3 are younger than the lower-intercept in Humayun et al.[36] but still consistent with error-bars, which are large, we have used upper-limit for Z2_1 and Z2_3 ages to determine their $\varphi_{loss}$. Given that Humayun et al.[36] and McCubbin et al.[17] obtained different ages for the lower-intercept of the general U–Pb Discordia built from Martian zircon data, we have further tested the influence of the age of the metamorphic event (i.e. 1700 or 1500 Ma), in which case we have recalculated $\varphi_{loss}$ using a 4430–1500 Ma Discordia. A summary of all parameters is given in Supplementary Data 5 and results are presented in Table 1. One could argue that some alteration could have also occurred at the age of the metamorphic event, except for zircon domains with $^{207}Pb/^{206}Pb$ ages consistent with original igneous crystallization. This is not an issue since the calculation of Pb-loss at 1500–1700 Ma is exactly the same as if U was increased. Indeed, lowering the Pb concentration by a factor of 0.2 (80% Pb-loss) is equivalent to increasing the concentration of U by a factor of 5. However, our Pb-loss term assumes no Th/U increase. If this happened, it would mean that the enrichment factor used in our calculations would have to be decreased, which would ultimately result in younger ages than those determined. However, if an enrichment occurred at 1500–1700 Ma it must have been limited to account for the preservation of the decoupling between measured and time-integrated Th/U. Finally, whether or not alteration was recorded after the metamorphic event would have no influence on the determined ages because the lower bound is defined by zircon domains that were not affected by the metamorphic event.

## Data availability

All data used in this study are available in the Supplementary Data published alongside this manuscript.

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

## Acknowledgements

M.G. acknowledges financial support from Matthias Willbold and NERC through grant NE/J018031/2, as well as LabEx ClerVolc (ANR-10-LABX-0006), Région Auvergne (Auvergne Fellowship program), and the French Agence Nationale de la Recherche (ANR) through the project *Zircontinents* (ANR-17-CE31-0021). This is Laboratory of Excellence ClerVolc contribution 342. J.F. is supported by a CNRS Momentum grant and CNRS-INSU through the Programme National de Planétologie.

## Author contributions

M.G. designed the research, compiled the data, did the calculations and modeling, and interpreted the data. M.G. and J.F. discussed the results and their implications, and wrote the manuscript.

## Additional information

**Competing interests:** The authors declare no competing interests.

