## [Peer Review File · Nature Communications]

Reviewers' comments:

Reviewer #1 (Remarks to the Author):

The authors employ a modeling approach to assess secondary processes that alter zircon U/Th/Pb schematics. By applying the model validated from the literature data of terrestrial and lunar zircons, authors applied this model to the literature data of zircons in Martian meteorite NWA 7034/7533. Results show some zircons in NWA 7034/7533 show low-temperature aqueous alteration. The methodology is intriguing and original, and the discussion and conclusion are reasonable supported by the methods and results. I do not see any major issues with the manuscript, and thus I recommend acceptance after minor revisions.

There are several comments that may help to make the paper clearer:

- 1) Authors did not get to the Martian zircons until 3 pages into the main text. It would be better to add a few introductory sentences at the beginning of the main text.
- 2) It is unclear how authors derived the alteration age of zircons to be between 42 and 564 Ma? These ages appeared rather suddenly on line 132 to 135. Since this is the crux of the results, please explain how the ages are calculated.
- 3) For the zircons showing the aqueous alteration in Figure 3, authors should give the sample numbers so that the readers can trace the images shown in the original papers. This is especially useful considering textural evidence was provided by Liu et al. (2015) about hydrothermal alterations of zircons in these samples (see reference below).
- 4) Natural communication allows more pages. It would be helpful for authors to move some supplementary materials to the main text, so that the outlines of how

Minor comments:

Line 19: should be "Evidence exists that supports the "

References

1. Y. Liu, C. Ma, J. Beckett (2015c). Hydrothermal alteration of martian zircons in NWA 7034/7533. 78th MetSoc, #5080.

Reviewer #2 (Remarks to the Author):

Review of

Record of low-temperature aqueous alteration of Martiam zircon during the late Amazonian

By Martin Guitreau and Jessica Flauhaut

Assessment

Guitreau and Flauhaut propose that anomalous Th/U ratios in some zircons from a Martian meteorite can be explained by interaction with weathering solutions in much the same way as anomalous Th/U in ancient zircons from the Jack

However the explanation of these authors of a decoupling of the Th-U-Pb system of the Martian zircons as a result of weathering is very different from the mechanism proposed for the weathering of the Jack Hills zircon by Pidgeon et al.. Weathering of the Jack Hills zircons involves transport of U and Th and other elements present in surrounding weathering fluids into the radiation damaged zircons via cracks and possibly by percolation into highly metamict domains. This results in an increase in the U and Th concentrations and the measured Th/U. U-Th-Pb discordance is interpreted as the result of U and Th gain not Pb loss and any loss of radiogenic Pb is viewed as insignificant.

In contrast the Guiteeau and Flauhaut model proposes that the anomalous Th/U ratios are due to decoupling of the U-Th-Pb system in the Martian zircon by low temperature alteration by aqueous solutions and attribute discordance of the Martian zircons to loss of radiogenic Pb. In this model the Th and U are rearranged within the zircon. There is no mention in the model of gain of U and Th from the environmental solutions so, as far as the present reviewer can see, the two models are in no way similar and there is no basis, as suggested in the paper, to draw parallels between the two models.

Consequently the present reviewer cannot recommend the paper for publication in its present form. It could well be that the anomalous Th/U in Martian zircons is the result of interaction with the hydrosphere along the lines of the fluid catalyzed, decoupling - Pb loss model. It is up to the authors to provide convincing evidence in support of this model. But their decoupling model is nothing like the Jack Hills zircon weathering model, which involves addition of U and Th from weathering solutions without significant loss of radiogenic Pb, and in any revision of the paper the differences between the two models should be pointed out or the JH weathering paper could be mentioned in passing or not at all, with the revision focusing on a description of the decoupling model as a new stand-alone concept.

Some specific comments

Line 41. The oldest terrestrial zircon is ~ 4375Ma. The reviewer has no idea how the authors extracted an age of 4438 Ma from the reference Valley paper .

Line 44-45. "Radiation damage accumulates"

Line 47. Can be mobilized

line 48. Remove “ aging”:

line 50. As far as the reviewer is aware references 17-19 do not deal with oxygen isotopes

Line 52 What are advanced signs of radiation damage?

(line 68 -71 and also line 72-77) . These sentences misrepresent what is described in the Pidgeon et al. paper. They interpret the Pidgeon et al. paper as saying that the anomalous Th/U ratios are associated with areas of high radiation damage in the zircon and are the result of “decoupling” of Th from U. However, it is reported in the Pidgeon et al. paper that anomalous Th/U, enhanced U and Th concentration, elevated $\delta^{18}\text{O}$ and common Pb , anomalous OH and U-Th-Pb zero age discordance are found in SIMS analyses overlapping cracks and are the result of materials including water being added to the zircon during penetration of aqueous fluids into cracks and metamict parts. It is not correct to attribute the anomalous Th/U results of the Jack Hills to decoupling. This term is not mentioned in the Pidgeon et al. paper .

Also, The present authors deal only with the Th/U ratios. The Pidgeon et al. paper looks at a combination of parameters besides Th/U , including U and Th contents, Th-U-Pb discordance, common Pb content, $\delta^{18}\text{O}$, and location of SIMS analysis on the grain – e.g. whether the SIMS analysis overlaps a crack or a zone. The present authors would need to review these additional parameters (where the data are available) to properly assess the likelihood that the Jack hills zircon low temperature fluid interaction mechanism could have affected the Martian zircons.

line 68 73 and elsewhere. Decoupling is the fundamental mechanism referred to in the Guitreau Flahaut paper and the term “ decoupling “: needs to be defined. Does it refer to independent movement of U-Th and Pb within a zircon during low temperature fluid interaction? Is the zircon viewed as a closed system to U, Th and Pb? Or is it envisaged that radiogenic Pb can escape during weathering interaction?. What about U and Th ? Is it envisaged that these elements are contained within the zircon or can these also be leached out of the zircon during low temperature aqueous interaction?

Line 72 to 75.

This raises the issue of the α -dose shown Fig2 It should be kept in mind that zircons can be annealed in nature and it has been shown that for the JH zircons the α -dose determined from the 207/206 age and the measured Th and U is a gross overestimate of the dose needed to produce the observed radiation damage in the zircon. This is referred to briefly in your method section.

Line 82. The sentence “Lead mobility which accounts for most cases of U-Pb discordance”. contradicts the conclusions of the Pidgeon et al. paper that discordance of the Jack Hills zircons is due to U and Th gain. There are a number of hydrothermal experiments that demonstrate episodic loss of radiogenic Pb from radiation damaged zircon . These experiments result in annealing of the zircon. In low temperature weathering it is envisaged that solutions infiltrate the zircon mainly through cracks but also into radiation damaged zones (to be demonstrated) where they precipitate trace elements including U and Th but cause no annealing of the radiation damage and little if any

loss of radiogenic Pb .(except possibly in some damaged zones which is yet to be demonstrated). Essentially all the U-Th-Pb discordance is the result of U and Th gain and not Pb loss.

Line 86 – 104 Presents arguments for limited Th/U variation in natural systems.

Line 96. This process would probably also remove much of the U and Th

Line 97-100. With the possible exception of ICPMS data, Pb-Pb and U-Th-Pb ratios are routinely corrected for common Pb using 204

Line 105 the sentence starting. “ Available data presented in supplementary material (Fig.2S, Table S3) etc. Is one of a number of examples of the use of undefined terms such as “ first percolation point” (used elsewhere) and unexplained statements such as “ expected from current knowledge of lunar surface evolution” and would be incomprehensible to anyone not totally familiar with radiation damage and lunar literature

105 and on.

This is difficult to review as on the one hand it interprets the anomalous Th/U as evidence low temperature alteration “ much as what was experienced by crystals from Western Australia “ and on the other applies the data to “constrain the age of decoupling between Th/U and Pb isotopes”, which is not what was experienced by crystals from Western Australia. With this in mind the reviewer has not extended the review into the section which sets out to determine the age of weathering.

Reviewer #3 (Remarks to the Author):

Review of “Record of low-temperature aqueous alteration of Martian zircon during the late Amazonian” by Guitreau and Flahaut.

The timing of aqueous alteration on Mars is of fundamental importance in planetary science and astrobiology. The present study investigated this issue through analysis of previously published U-Th-Pb isotope data for zircon in two ancient Martian meteorites. Specifically, the authors constrain the timing of secondary Th/U fractionation in Martian zircon by comparing measured Th/U with “time-integrated Th/U” calculated from measured $^{208}\text{Pb}/^{206}\text{Pb}$ (i.e., ratio of ^{232}Th and ^{238}U daughter nuclides), given that such secondary Th/U fractionation has been documented in aqueously altered terrestrial zircon (Pidgeon et al., 2017). The analytical results revealed that Th/U fractionation has occurred in geologically young ages within one Martian zircon grain (NWA 7533-Z2). Assuming that the zircon grain initially had a Th/U ratio between 0.5 and 1.0, the timing of Th/U

fractionation was estimated as between 564 and 42 Ma. This led the authors to argue that aqueous alteration should occur during the late Amazonian on Mars.

The subject addressed in the present study is of wide interest and the approach is novel. I am afraid, however, that the secondary Th/U fractionation observed in zircon NWA 75333-Z2 may be due to recent aqueous alteration on the Earth rather than Mars. Considering ample evidence of terrestrial alteration for the Martian meteorite (Lorand et al., 2015), this possibility should be critically examined before the manuscript is acceptable for publication. The possibility might be excluded if the minimum age estimate of 42 Ma for the Th/U fractionation was robust. Unfortunately, however, the age estimate depends highly on the assumed initial Th/U ratio in the zircon grain: the age of secondary Th/U fractionation can be down to ~0 Ma when the initial Th/U ratio is assumed to be ~2.0, which is well within the range of measured Th/U ratios in other Martian zircon grains that have not undergone recent Th/U fractionation (Fig. 3). Alternatively, positive $\Delta^{17}\text{O}$ values obtained from zircon NWA 75333-Z2 (Nemchin et al., 2014) may be taken as evidence for alteration on Mars. Please note, however, that the elevated values could be analytical artifacts due to increased $^{16}\text{OH}/^{16}\text{O}$ ratios, as observed for altered terrestrial zircon grains (Pidgeon et al., 2017). Thus, I feel that the conclusion of late Amazonian alteration on Mars is still unsound.

Additional comments:

Solubility of U

Th/U fractionation during aqueous alteration is a result of the higher solubility of U in fluid than Th. Note, however, that the U solubility is dependent on pH and $f\text{O}_2$ conditions. For instance, it is well known that U has been essentially insoluble under reduced surface conditions on the Archean Earth. Thus, it needs to be evaluated whether Th/U fractionation is viable during aqueous alteration under Mars surface conditions.

Radiation damage or fracture

It is certainly true that radiation damage is an important factor for controlling the effect of secondary alteration on zircon chemistry. In the case of Martian zircon NWA 75333-Z2, however, fractures in analytical spots rather than radiation damage seem to be the main cause for the elevated measured Th/U ratios (see images of the grain presented in supplementary information of Nemchin et al., 2014). Indeed, Pidgeon et al. (2017) demonstrated that analyses which overlap a fracture yield highly elevated Th contents and Th/U ratios, corresponding well with the data for NWA 75333-Z2.

Response to reviewers

Note: line numbers given in our responses referred to the final version of the manuscript (with track-changes accepted).

Reviewer #1 comments:

1) Authors did not get to the Martian zircons until 3 pages into the main text. It would be better to add a few introductory sentences at the beginning of the main text.

Response: As suggested, we added some text to introduce Martian zircons lines 51-55, just after describing the zircon mineral specificities.

2) It is unclear how authors derived the alteration age of zircons to be between 42 and 564 Ma? These ages appeared rather suddenly on line 132 to 135. Since this is the crux of the results, please explain how the ages are calculated.

Response: We have further developed explanations about our model and its results at lines 178-217 to make our point more explicit. In addition, we have modified the caption to Figure 5 (where the time-integrated Th/U can be related to the alteration age) to assist the reader in understanding the results of our two-stage model. In addition, results for the three-stage model are now presented as a table (Table 1) for clarity.

3) For the zircons showing the aqueous alteration in Figure 3, authors should give the sample numbers so that the readers can trace the images shown in the original papers. This is especially useful considering textural evidence was provided by Liu et al. (2015) about hydrothermal alterations of zircons in these samples (see reference below).

Response: As suggested, sample numbers were inserted next to the data points.

4) Natural communication allows more pages. It would be helpful for authors to move some supplementary materials to the main text, so that the outlines of how

Response: Thank you for this suggestion. As proposed, we have increased the text length to add some key details about the model within the main body of the article. We believe it makes the manuscript much clearer and allowed us to address the comments of the second and third reviewers at the same time (see below).

Minor comments:

Line 19: should be “Evidence exists that supports the ”

Response: Thank you for spotting this. We changed the text accordingly.

References

1. Y. Liu, C. Ma, J. Beckett (2015c). Hydrothermal alteration of martian zircons in NWA 7034/7533. 78th MetSoc, #5080.

Response: We have added this reference as suggested.

Reviewer #2 comments:

Guitreau and Flahaut propose that anomalous Th/U ratios in some zircons from a Martian meteorite can be explained by interaction with weathering solutions in much the same way as anomalous Th/U in ancient zircons from the Jack.

However the explanation of these authors of a decoupling of the Th-U-Pb system of the Martian zircons as a result of weathering is very different from the mechanism proposed for the weathering of the Jack Hills zircon by Pidgeon et al.. Weathering of the Jack Hills zircons involves transport of U and Th and other elements present in surrounding weathering fluids into the radiation damaged zircons via cracks and possibly by percolation into highly metamict domains. This results in an increase in the U and Th concentrations and the measured Th/U. U-Th-Pb discordance is interpreted as the result of U and Th gain not Pb loss and any loss of radiogenic Pb is viewed as insignificant.

In contrast the Guitreau and Flahaut model proposes that the anomalous Th/U ratios are due to decoupling of the U-Th-Pb system in the Martian zircon by low temperature alteration by aqueous solutions and attribute discordance of the Martian zircons to loss of radiogenic Pb. In this model the Th and U are rearranged within the zircon. There is no mention in the model of gain of U and Th from the environmental solutions so, as far as the present reviewer can see, the two models are in no way similar and there is no basis, as suggested in the paper, to draw parallels between the two models.

Consequently the present reviewer cannot recommend the paper for publication in its present form. It could well be that the anomalous Th/U in Martian zircons is the result of interaction with the hydrosphere along the lines of the fluid catalyzed, decoupling - Pb loss model. It is up to the authors to provide convincing evidence in support of this model. But their decoupling model is nothing like the Jack Hills zircon weathering model, which involves addition of U and Th from weathering solutions without significant loss of radiogenic Pb, and in any revision of the paper the differences between the two models should be pointed out or the JH weathering paper could be mentioned in passing or not at all, with the revision focusing on a description of the decoupling model as a new stand-alone concept.

Response: We completely agree with the reviewer about Jack Hills zircons. In fact, we realize that the reviewer misinterpreted our message, because nowhere in the main text it was written that we interpreted Pidgeon et al. paper to say that Pb-loss is the cause for U-Pb discordance in Jack Hills zircons. However, it is true that the wording in the method section may have been misleading and we apologize for this issue. We have rewritten the text in a more explicit manner and have defined terms that the reviewer was confused about.

Line 41. The oldest terrestrial zircon is ~ 4375Ma. The reviewer has no idea how the authors extracted an age of 4438 Ma from the reference Valley paper.

Response: This was a typo. We meant to write 4378 Ma. Corrected.

Line 44-45. “Radiation damage accumulates”

Response: Changed accordingly.

Line 47. Can be mobilized

Response: Changed accordingly.

Line 48. Remove “ aging”:

Response: Done.

Line 50. As far as the reviewer is aware references 17-19 do not deal with oxygen isotopes

Response: We apologize for this mistake. We have added the proper references that include O isotope measurements.

Line 52 What are advanced signs of radiation damage?

Response: We have explained what we meant by advanced signs of radiation damage at lines 45-47 and 60-66.

Line 68 -71 and also line 72-77. These sentences misrepresent what is described in the Pidgeon et al. paper. They interpret the Pidgeon et al. paper as saying that the anomalous Th/U ratios are associated with areas of high radiation damage in the zircon and are the result of “decoupling” of Th from U. However, it is reported in the Pidgeon et al. paper that anomalous Th/U, enhanced U and Th concentration, elevated $\delta^{18}\text{O}$ and common Pb, anomalous OH and U-Th-Pb zero age discordance are found in SIMS analyses overlapping cracks and are the result of materials including water being added to the zircon during penetration of aqueous fluids into cracks and metamict parts. It is not correct to attribute the anomalous Th/U results of the Jack Hills to decoupling. This term is not mentioned in the Pidgeon et al. paper .

Also, The present authors deal only with the Th/U ratios. The Pidgeon et al. paper looks at a combination of parameters besides Th/U, including U and Th contents, Th-U-Pb discordance, common Pb content, $\delta^{18}\text{O}$, and location of SIMS analysis on the grain – e.g. whether the SIMS analysis overlaps a crack or a zone. The present authors would need to review these additional parameters (where the data are available) to properly assess the likelihood that the Jack hills zircon low temperature fluid interaction mechanism could have affected the Martian zircons.

Response: As stated above, our initial description was unclear and probably clumsy, as our message was not understood. We have made significant changes to the text, added a clear description of the model, and defined the term “decoupling “ (which is in the end, the discrepancy between the measured Th/U and the time-integrated Th/U). We totally agree with Pidgeon et al. conclusions about Jack Hills zircons and we are sorry if they were not well reported in our initial version of the text. We have significantly modified the text so as to accurately represent Pidgeon et al. conclusions.

Line 68-73 and elsewhere. Decoupling is the fundamental mechanism referred to in the Guitreau Flahaut paper and the term “decoupling “: needs to be defined. Does it refer to independent movement of U-Th and Pb within a zircon during low temperature fluid interaction? Is the zircon viewed as a closed system to U, Th and Pb? Or is it envisaged that radiogenic Pb can escape during weathering interaction? What about U and Th? Is it envisaged that these elements are contained within the zircon or can these also be leached out of the zircon during low temperature aqueous interaction?

Response: In the present manuscript, decoupling is not used to describe a mechanism but only used as a descriptive term. It merely illustrates the fact that measured and time-integrated Th/U ratios are different, no matter which mechanism or process is responsible for it. We have made this definition more explicit in the text (l.83-84).

Line 72 to 75. This raises the issue of the alpha-dose shown Fig2 It should be kept in mind that zircons can be annealed in nature and it has been shown that for the JH zircons the alpha-dose determined from the 207/206 age and the measured Th and U is a gross overestimate of the dose needed to produce the observed radiation damage in the zircon. This is referred to briefly in your method section.

Response: We have emphasized the inaccurate nature of radiation doses in the revised manuscript (e.g., l. 91-98).

Line 82. The sentence “Lead mobility which accounts for most cases of U-Pb discordance”. contradicts the conclusions of the Pidgeon et al. paper that discordance of the Jack Hills zircons is due to U and Th gain. There are a number of hydrothermal experiments that demonstrate episodic loss of radiogenic Pb from radiation damaged zircon. These experiments result in annealing of the zircon. In low temperature weathering it is envisaged that solutions infiltrate the zircon mainly through cracks but also into radiation damaged zones (to be demonstrated) where they precipitate trace elements including U and Th but cause no annealing of the radiation damage and little if any loss of radiogenic Pb .(except possibly in some damaged zones which is yet to be demonstrated). Essentially all the U-Th-Pb discordance is the result of U and Th gain and not Pb loss.

Response: In this sentence we were precisely referring to cases when zircon annealing occurred because we are discussing if Pb-loss could produce the decoupling we observe. This sentence was not at all referring to alteration such as seen by the Jack Hills zircons. We totally agree about the U and Th increase and have made this idea more explicit in the text.

Line 86 – 104 Presents arguments for limited Th/U variation in natural systems.

Response: We were only referring to observed Th/U in natural zircon. We have clarified this point.

Line 96. This process would probably also remove much of the U and Th

Response: We respectfully think different as U and Th substitute relatively well to Zr in the zircon lattice. Therefore, U and Th would still be present. It is true, though, that U and Th concentrations would be moderate.

Line 97-100. With the possible exception of ICPMS data, Pb-Pb and U-Th-Pb ratios are routinely corrected for common Pb using 204.

Response: We have added this idea in the revised text (l.133-134).

Line 105 the sentence starting. “ Available data presented in supplementary material (Fig.2S, Table S3) etc. Is one of a number of examples of the use of undefined terms such as “ first percolation point” (used elsewhere) and unexplained statements such as “expected from current knowledge of lunar surface evolution” and would be incomprehensible to anyone not totally familiar with radiation damage and lunar literature

Response: We have defined first percolation point (lines 43-48) and have tried our best to be more explicit in the text (e.g., we have inserted a couple of sentences on lunar zircons lines 140-144).

Line 105 and on. This is difficult to review as on the one hand it interprets the anomalous Th/U as evidence low temperature alteration “much as what was experienced by crystals from Western Australia “ and on the other applies the data to “constrain the age of decoupling between Th/U and Pb isotopes”, which is not what was experienced by crystals from Western Australia. With this in mind the reviewer has not extended the review into the section which sets out to determine the age of weathering.

Response: As stated before, we have made significant modifications to the main body of the text to clarify our model and interpretation of the observed “decoupling”, we are grateful for the comments which helped us rework the manuscript in a much clearer way and hope the reviewer is satisfied with these changes.

Reviewer #3 comments:

The subject addressed in the present study is of wide interest and the approach is novel. I am afraid, however, that the secondary Th/U fractionation observed in zircon NWA 75333-Z2 may be due to recent aqueous alteration on the Earth rather than Mars. Considering ample evidence of terrestrial alteration for the Martian meteorite (Lorand et al., 2015), this possibility should be critically examined before the manuscript is acceptable for publication. The possibility might be excluded if the minimum age estimate of 42 Ma for the Th/U fractionation was robust. Unfortunately, however, the age estimate depends highly on the assumed initial Th/U ratio in the zircon grain: the age of secondary Th/U fractionation can be down to ~0 Ma when the initial Th/U ratio is assumed to be ~2.0, which is well within the range of measured Th/U ratios in other Martian zircon grains that have not undergone recent Th/U fractionation (Fig. 3). Alternatively, positive $\Delta 17O$ values obtained from zircon NWA 75333-Z2 (Nemchin et al., 2014) may be taken as evidence for alteration on Mars.

Please note, however, that the elevated values could be analytical artifacts due to increased $^{16}\text{OH}/^{16}\text{O}$ ratios, as observed for altered terrestrial zircon grains (Pidgeon et al., 2017). Thus, I feel that the conclusion of late Amazonian alteration on Mars is still unsound.

Response: We agree that this is a very important issue to address. The question of whether alteration observed in some martian meteorites (e.g., precipitated carbonates or sulfates in shergottites) could be of terrestrial origin has been an ongoing debate for decades. However, we think that the reviewer's points are not a real issue in the case of our zircons, because the analyzed domains that show Th/U decoupling do not show anomalous $\Delta^{17}\text{O}$ values but rather regular mass-dependent fractionation along the Martian fractionation line. Therefore, the data are robust and not biased by analytical artifacts. Furthermore, assuming an initial Th/U ratio of 2 for the Martian zircons, as proposed by the reviewer, is simply impossible as the initial Th/U cannot be larger than the time-integrated one given that these crystals experienced an increase in Th/U ratio. Moreover, careful examination of textural and chemical characteristics of most well-behaved magmatic zircons indicate that their measured Th/U range between 0.4 and 0.8, which is very consistent with high-precision ID-TIMS measurements that gave an even narrower range of 0.6 to 0.7. This range is perfectly compatible with Th/U in terrestrial magmatic zircons, as well as in pristine lunar zircons. Therefore, we judge that a Th/U of 1, contrary to what the reviewer suggests, marks an absolute upper-limit for Martian magmatic zircons. Consequently, we respectfully think that comments made by reviewer 3 have no bearings on our study. Regarding the robustness of determined ages, we have refined our model so as to provide a more robust and more realistic estimate and we now think that our results are definitively conclusive.

Additional comments:

Solubility of U

Th/U fractionation during aqueous alteration is a result of the higher solubility of U in fluid than Th. Note, however, that the U solubility is dependent on pH and $f\text{O}_2$ conditions. For instance, it is well known that U has been essentially insoluble under reduced surface conditions on the Archean Earth. Thus, it needs to be evaluated whether Th/U fractionation is viable during aqueous alteration under Mars surface conditions.

Response: The reviewer raises an important point here. In fact, Martian conditions are oxidizing and therefore perfectly consistent with an alteration event like the one we envisage to happen on Mars. We have added some text discussing redox conditions on Mars and the possibility for U to be mobilized.

Radiation damage or fracture

It is certainly true that radiation damage is an important factor for controlling the effect of secondary alteration on zircon chemistry. In the case of Martian zircon NWA 7533 Z2, however, fractures in analytical spots rather than radiation damage seem to be the main cause for the elevated measured Th/U ratios (see images of the grain presented in supplementary information of Nemchin et al., 2014). Indeed, Pidgeon et al. (2017) demonstrated that analyses which overlap a fracture yield highly elevated Th contents and Th/U ratios, corresponding well with the data for NWA 7533-Z2.

Response: This is precisely the principle of this approach as weathering solutions penetrate within zircon through cracks and then propagate within amorphous domains. The point is that all authors who analyzed Jack Hills zircons observed this fact, whereas nobody reported this for ancient terrestrial igneous zircons. It is very unlikely that every single analysis avoided even little cracks. Yet, only Jack Hills zircons exhibit decoupling between measured and time-integrated Th/U. This decoupling was also seen using electron microprobe in Pidgeon et al. and not only when the beam overlapped a crack. In addition, not all analytical spots overlapped cracks in Nemchin et al. (2014) and some of the analyzed domains still provided the accurate age of 4430 Ma. Therefore, we are firmly convinced that these signals are real and cannot represent analytical artifacts.

Reviewers' comments:

Reviewer #3 (Remarks to the Author):

Review of "Record of low-temperature aqueous alteration of Martian zircon during the late Amazonian" by Guitreau and Flahaut

I found that the manuscript has been improved from the previous version: the possible effect of terrestrial weathering on zircon U/Th has been made (l. 240-249); the solubility of U under Martian surface conditions has been discussed (l. 235-240). These modifications strengthen the conclusion that the aqueous alteration occurred during the late Amazonian on Mars. I recommend publication of this manuscript in Nature Communication with relatively minor (but important) revision.

l. 35: 'thereby suggesting that Mars may still be habitable'—This seems to be an overstatement.

l. 98-110 & Fig. 2: I would suggest that calculated radiation doses in the Martian zircons are compared with those in the Jack Hills zircons. I am wondering if cracks in the Martian zircons are mainly due to impact rather than radiation damage accumulation. In fact, the secondary Th/U fractionation is observed in a grain showing relatively low U contents.

l. 204-205: 'much like what U-series nuclides do'—This is certainly not the case, as ^{206}Pb and ^{208}Pb are stable isotopes. Instead, the lack of decoupling, despite of the evidence of alteration, indicates either that Th/U fractionation during the alteration was negligible or that Th/U fractionation was accompanied by significant Pb-loss.

l. 209-210: 'We have used the same Th/U ratios ... as in the two-stage model'—For the two-stage model, it is reasonable to assume that the original Th/U ratio is within the range of the ratios observed in ancient well-behaved Martian zircons. But, the assumption would be untenable for the three-stage model, as some 1500-1700 Ma zircon grains have consistent time-integrated and measured Th/U ratios of up to ~ 1.8 (i.e., 1500-1700 Ma thermal event would cause Th/U elevation accompanied by significant Pb-loss). Thus, I am still considering that the timing of alteration responsible for the Th/U decoupling could be significantly younger than 56 Ma, though I agree with the authors that the $\Delta 17\text{O}$ data suggest that the alteration would be on Mars rather than Earth.

l. 240-241: 'Although the timing does not match'—See above comment.

l. 244-245: 'O isotope data for Z2 zircon ..'—It would be better to emphasize that the two analytical spots showing the highest measured Th/U ratios yielded the highest $\Delta 17\text{O}$ values within the grain and also that the $\Delta 17\text{O}$ values are coincident with that of carbonates from ALH 84001 (Nemchin et al., 2014).

l. 245-249: Note that neither calcite nor barite veins exist in the host rocks of the Jack Hills zircons showing secondary Th/U fractionation.

Response to reviewers for the second round of revisions

Reviewer #3 comments:

Review of “Record of low-temperature aqueous alteration of Martian zircon during the late Amazonian” by Guitreau and Flahaut.

I found that the manuscript has been improved from the previous version: the possible effect of terrestrial weathering on zircon U/Th has been made (l. 240-249); the solubility of U under Martian surface conditions has been discussed (l. 235-240). These modifications strengthen the conclusion that the aqueous alteration occurred during the late Amazonian on Mars. I recommend publication of this manuscript in Nature Communication with relatively minor (but important) revision.

l. 35: ‘thereby suggesting that Mars may still be habitable’—This seems to be an overstatement.

Response: In order to comply with the reviewer’s comment, we have changed “may” for “might” so as not to be too assertive.

l. 98-110 & Fig. 2: I would suggest that calculated radiation doses in the Martian zircons are compared with those in the Jack Hills zircons. I am wondering if cracks in the Martian zircons are mainly due to impact rather than radiation damage accumulation. In fact, the secondary Th/U fractionation is observed in a grain showing relatively low U contents.

Response: Data are already available in Table S5 and show values beyond the first percolation point. Moreover, zircon textures revealed by cathodoluminescence and back-scattered electron images are consistent with zircons that underwent advanced radiation damage as we already state in the main text (e.g., Liu et al., 2015 Hydrothermal alteration of martian zircons in NWA 7034/7533. 78th MetSoc, #5080). However, whether or not cracks are radiation-damage-related, they do represent pathways for fluid infiltration and weathering.

l. 204-205: ‘much like what U-series nuclides do’—This is certainly not the case, as ^{206}Pb and ^{208}Pb are stable isotopes. Instead, the lack of decoupling, despite of the evidence of alteration, indicates either that Th/U fractionation during the alteration was negligible or that Th/U fractionation was accompanied by significant Pb-loss.

Response: We respectfully think that the reviewer missed the point here. We, of course, acknowledge, that ^{206}Pb and ^{208}Pb are stable isotopes that are at the end of the radioactive decay chains of ^{238}U and ^{232}Th . The point with radioactive nuclides of the U-series is that the entire series goes back to secular equilibrium, hence, from U down to Pb. In the text, we merely explain that radiogenic accumulation of ^{208}Pb and ^{206}Pb by ^{232}Th and ^{238}U decay will progressively erase the decoupling until it reaches a secular equilibrium where the $^{208}\text{Pb}/^{206}\text{Pb}$ ratio will reflect the $^{232}\text{Th}/^{238}\text{U}$ ratio.

l. 209-210: ‘We have used the same Th/U ratios ... as in the two-stage model’—For

the two-stage model, it is reasonable to assume that the original Th/U ratio is within the range of the ratios observed in ancient well-behaved Martian zircons. But, the assumption would be untenable for the three-stage model, as some 1500-1700 Ma zircon grains have consistent time-integrated and measured Th/U ratios of up to ~1.8 (i.e., 1500-1700 Ma thermal event would cause Th/U elevation accompanied by significant Pb-loss). Thus, I am still considering that the timing of alteration responsible for the Th/U decoupling could be significantly younger than 56 Ma, though I agree with the authors that the $\Delta^{17}\text{O}$ data suggest that the alteration would be on Mars rather than Earth.

Response: We agree that alteration and Th/U modification could have occurred at 1500-1700 Ma following metamorphism. As a matter of fact, we already discussed this possibility in the Method section (lines 117-129) where we wrote: “One could argue that some alteration could have also occurred at the age of the metamorphic event, except for zircon domains with $^{207}\text{Pb}/^{206}\text{Pb}$ ages consistent original igneous crystallization. This is not an issue since the calculation of Pb-loss at 1500-1700 Ma is exactly the same as if U was increased. Indeed, lowering the Pb concentration by a factor of 0.2 (80% Pb-loss) is equivalent to increasing the concentration of U by a factor of 5. However, our Pb-loss term assumes no Th/U increase. If this happened, it would mean that the enrichment factor used in our calculations would have to be decreased, which would ultimately result in younger ages than those determined. However, if an enrichment occurred at 1500-1700 Ma it must have been limited to account for the preservation of the decoupling between measured and time-integrated Th/U. Consequently, whether or not alteration was recorded after the metamorphic event would have no influence on the final results as the lower bound is determined by zircon domains that were not affected by the metamorphic event”. Therefore, modeling the alteration age with an arbitrarily set value for pre-alteration Th/U would artificially decrease the maximum age estimate. In the absence of evidence for Th/U modification at 1500-1700 Ma in zircons displaying decoupling, we purposely kept the values for no-modification at 1500-1700 Ma so as to widen the possible scenarios and, in turn, obtain least biased estimates. Finally, taking into consideration modification of Th/U at 1500-1700 Ma would not result in age estimates younger than 56 Ma.

l. 240-241: ‘Although the timing does not match’—See above comment.

Response: See above response.

l. 244-245: ‘O isotope data for Z2 zircon ..’—It would be better to emphasize that the two analytical spots showing the highest measured Th/U ratios yielded the highest $\Delta^{17}\text{O}$ values within the grain and also that the $\Delta^{17}\text{O}$ values are coincident with that of carbonates from ALH 84001 (Nemchin et al., 2014).

Response: We have mentioned in the main text that the O isotope composition of Z2 zircons shared similarities with carbonates in ALH 84001, together with the fact that they align along the Martian fractionation line.

l. 245-249: Note that neither calcite nor barite veins exist in the host rocks of the Jack Hills zircons showing secondary Th/U fractionation.

Response: We respectfully think that this comment is off topic and that there is a misunderstanding by Reviewer 3. When referred to barite and calcite veins, we were talking about terrestrial alteration in hot deserts that experienced Martian meteorites. Why would Jack Hills zircon host-rocks (conglomerates and quartzites) have to also have this kind of veins and, on top of that, show a decoupling in these very veins? They have not been found in hot deserts and they are terrestrial sedimentary rocks that have experienced intense weathering on Earth since at least the Permian.